# Enabling Deep Spiking Neural Networks with Hybrid Conversion and Spike Timing Dependent Backpropagation

**Nitin Rathi[1], Gopalakrishnan Srinivasan[1], Priyadarshini Panda[2] & Kaushik Roy[1]**
[1]School of Electrical and Computer Engineering, Purdue University
[2]Department of Electrical Engineering, Yale University
`{rathi2, srinivg}@purdue.edu`, `priya.panda@yale.edu`,
` kaushik@purdue.edu`

## Abstract

Spiking Neural Networks (SNNs) operate with asynchronous discrete events (or spikes) which can potentially lead to higher energy-efficiency in neuromorphic hardware implementations. Many works have shown that an SNN for inference can be formed by copying the weights from a trained Artificial Neural Network (ANN) and setting the firing threshold for each layer as the maximum input received in that layer. These type of converted SNNs require a large number of time steps to achieve competitive accuracy which diminishes the energy savings. The number of time steps can be reduced by training SNNs with spike-based backpropagation from scratch, but that is computationally expensive and slow. To address these challenges, we present a computationally-efficient training technique for deep SNNs[1]. We propose a hybrid training methodology: 1) take a converted SNN and use its weights and thresholds as an initialization step for spike-based backpropagation, and 2) perform incremental spike-timing dependent backpropagation (STDB) on this carefully initialized network to obtain an SNN that converges within few epochs and requires fewer time steps for input processing. STDB is performed with a novel surrogate gradient function defined using neuron's spike time. The weight update is proportional to the difference in spike timing between the current time step and the most recent time step the neuron generated an output spike. The SNNs trained with our hybrid conversion-and-STDB training perform at $10\times-25\times$ fewer number of time steps and achieve similar accuracy compared to purely converted SNNs. The proposed training methodology converges in less than 20 epochs of spike-based backpropagation for most standard image classification datasets, thereby greatly reducing the training complexity compared to training SNNs from scratch. We perform experiments on CIFAR-10, CIFAR-100 and ImageNet datasets for both VGG and ResNet architectures. We achieve top-1 accuracy of 65.19% for ImageNet dataset on SNN with 250 time steps, which is $10\times$ faster compared to converted SNNs with similar accuracy.

## 1 Introduction

In recent years, Spiking Neural Networks (SNNs) have shown promise towards enabling low-power machine intelligence with event-driven neuromorphic hardware. Founded on bio-plausibility, the neurons in an SNN compute and communicate information through discrete binary events (or spikes) a significant shift from the standard artificial neural networks (ANNs), which process data in a real-valued (or analog) manner. The binary all-or-nothing spike-based communication combined with sparse temporal processing precisely make SNNs a low-power alternative to conventional ANNs. With all its appeal for power efficiency, training SNNs still remains a challenge. The discontinuous and non-differentiable nature of a spiking neuron (generally, modeled

---

[1]https://github.com/nitin-rathi/hybrid-snn-conversion

as leaky-integrate-and-fire (LIF), or integrate-and-fire (IF)) poses difficulty to conduct gradient descent based backpropagation. Practically, SNNs still lag behind ANNs, in terms of performance or accuracy, in traditional learning tasks. Consequently, there has been several works over the past few years that propose different learning algorithms or learning rules for implementing deep convolutional SNNs for complex visual recognition tasks (Wu et al., 2019; Hunsberger & Eliasmith, 2015; Cao et al., 2015). Of all the techniques, conversion from ANN-to-SNN (Diehl et al., 2016; 2015; Sengupta et al., 2019; Hunsberger & Eliasmith, 2015) has yielded state-of-the-art accuracies matching deep ANN performance for Imagenet dataset on complex architectures (such as, VGG (Simonyan & Zisserman, 2014) and ResNet (He et al., 2016) ). In conversion, we train an ANN with ReLU neurons using gradient descent and then convert the ANN to an SNN with IF neurons by using suitable threshold balancing (Sengupta et al., 2019). But, SNNs obtained through conversion incur large latency of $2000-2500$ time steps (measured as total number of time steps required to process a given input image[2]). The term 'time step' defines an unit of time required to process a single input spike across all layers and represents the network latency. The large latency translates to higher energy consumption during inference, thereby, diminishing the efficiency improvements of SNNs over ANNs. To reduce the latency, spike-based backpropagation rules have been proposed that perform end-to-end gradient descent training on spike data. In spike-based backpropagation methods, the non-differentiability of the spiking neuron is handled by either approximating the spiking neuron model as continuous and differentiable (Huh & Sejnowski, 2018) or by defining a surrogate gradient as a continuous approximation of the real gradient (Wu et al., 2018; Bellec et al., 2018; Neftci et al., 2019). Spike-based SNN training reduces the overall latency by $\sim 10\times$ (for instance, $200-250$ time steps required to process an input (Lee et al., 2019)) but requires more training effort (in terms of total training iterations) than conversion approaches. A single feed-forward pass in ANN corresponds to multiple forward passes in SNN which is proportional to the number of time steps. In spike-based backpropagation, the backward pass requires the gradients to be integrated over the total number of time steps that increases the computation and memory complexity. The multiple-iteration training effort with exploding memory requirement (for backward pass computations) has limited the applicability of spike-based backpropagation methods to small datasets (like CIFAR10) on simple few-layered convolutional architectures.

In this work, we propose a hybrid training technique which combines ANN-SNN conversion and spike-based backpropagation that reduces the overall latency as well as decreases the training effort for convergence. We use ANN-SNN conversion as an initialization step followed by spike-based backpropagation incremental training (that converges to optimal accuracy with few epochs due to the precursory initialization). Essentially, our hybrid approach of taking a converted SNN and incrementally training it using backpropagation yields improved energy-efficiency as well as higher accuracy than a model trained from scratch with only conversion or only spike-based backpropagation.

In summary, this paper makes the following contributions:

- We introduce a hybrid computationally-efficient training methodology for deep SNNs. We use the weights and firing thresholds of an SNN converted from an ANN as the initialization step for spike-based backpropagation. We then train this initialized network with spike-based backpropagation for few epochs to perform inference at a reduced latency or time steps.

- We propose a novel spike time-dependent backpropagation (STDB, a variant of standard spike-based backpropagation) that computes surrogate gradient using neuron's spike time. The parameter update is triggered by the occurrence of spike and the gradient is computed based on the time difference between the current time step and the most recent time step the neuron generated an output spike. This is motivated from the Hebb's principle which states that the plasticity of a synapse is dependent on the spiking activity of the neurons connected to the synapse.

- Our hybrid approach with the novel surrogate gradient descent allows training of large-scale SNNs without exploding memory required during spike-based backpropagation. We evaluate our hybrid approach on large SNNs (VGG, ResNet-like architectures) on

---

[2]SNNs process Poisson rate-coded input spike trains, wherein, each pixel in an image is converted to a Poisson-distribution based spike train with the spiking frequency proportional to the pixel value

Imagenet, CIFAR datasets and show near iso-accuracy compared to similar ANNs and converted SNNs at lower compute cost and energy.

## 2 SPIKE TIMING DEPENDENT BACKPROPAGATION (STDB)

In this section, we describe the spiking neuron model, derive the equations for the proposed surrogate gradient based learning, present the weight initialization method for SNN, discuss the constraints applied for ANN-SNN conversion, and summarize the overall training methodology.

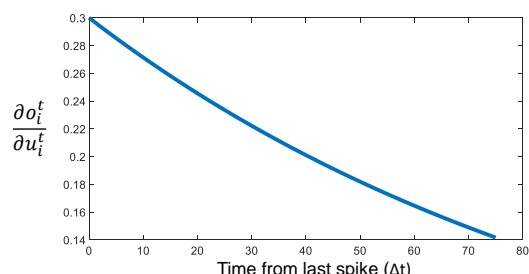

### 2.1 LEAKY INTEGRATE AND FIRE (LIF) NEURON MODEL

The neuron model defines the dynamics of the neuron's internal state and the trigger for it to generate a spike. The differential equation

$$\tau \frac{dU}{dt} = -(U - U_{rest}) + RI \qquad (1)$$

is widely used to characterize the leaky-integrate-and-fire (LIF) neuron model where, $U$ is the internal state of the neuron referred as the membrane potential, $U_{rest}$ is the resting potential, $R$ and $I$ are the input resistance and the current, respectively. The above equation is valid when the membrane potential is below the threshold value ($V$). The neuron geneartes an output spike when $U \geqslant V$

Figure 1: Surrogate gradient of the spiking neuron activation function (Eq. 11). $\alpha = 0.3, \beta = 0.01$. The gradient is computed for each neuron and $\Delta t$ defines the time difference between current simulation time and the last spike time of the neuron. For example, if a neuron spikes at $t_s = 12$ its gradient will be maximum at $t = 12(\Delta t = 0)$ and gradually decrease for later time steps. If the same neuron spikes later at $t_s = 24$ its previous spike history will be overwritten and the gradient computation for $t = 24$ onward will only consider the most recent spike. This avoids the overhead of storing all the spike history in memory.

and $U$ is reduced to the reset potential. This representation is described in continuous domain and more suitable for biological simulations. We modify the equation to be evaluated in a discrete manner in the Pytorch framework (Wu et al., 2018). The iterative model for a single post-neuron is described by

$$u_i^t = \lambda u_i^{t-1} + \sum_j w_{ij} o_j^t - v o_i^{t-1} \qquad (2)$$

$$o_i^{t-1} = \begin{cases} 1, & \text{if } u_i^{t-1} > v \\ 0, & \text{otherwise} \end{cases} \qquad (3)$$

where $u$ is the membrane potential, subscript $i$ and $j$ represent the post- and pre-neuron, respectively, superscript $t$ is the time step, $\lambda$ is a constant ($< 1$) responsible for the leak in membrane potential, $w$ is the weight connecting the pre- and post-neuron, $o$ is the binary output spike, and $v$ is the firing threshold potential. The right hand side of Equation 2 has three terms: the first term calculates the leak in the membrane potential from the previous time step, the second term integrates the input from the previous layer and adds it to the membrane potential, and the third term which is outside the summation reduces the membrane potential by the threshold value if a spike is generated. This is known as soft reset as the membrane potential is lowered by $v$ compared to hard reset where the membrane potential is reduced to the reset value. Soft reset enables the spiking neuron to carry forward the excess potential above the firing threshold to the following time step, thereby minimizing information loss.

**Algorithm 1** ANN-SNN conversion: initialization of weights and threshold voltages

---

**Input:** Trained ANN model ($A$), SNN model ($N$), Input ($X$)
// Copy ann weights to snn
  **for** *l=1* **to** *L* **do**
  |   $N_l.W \leftarrow A_l.W$
**end**
// Initialize threshold voltage to 0
  $V \leftarrow [0, \cdots, 0]_{L-1}$
  **for** *l=1* **to** *L-1* **do**
  |   $v \leftarrow 0$
  |   **for** *t=1* **to** *T* **do**
  |   |   $O_0^t \leftarrow PoissonGenerator(X)$
  |   |   **for** *k=1* **to** *l* **do**
  |   |   |   **if** $k < l$ **then**
  |   |   |   |   // Forward (Algorithm 3)
  |   |   |   **end**
  |   |   |   **else**
  |   |   |   |   // Pre-nonlinearity ($A$)
  |   |   |   |   $A \leftarrow N_l(O_{k-1}^t)$
  |   |   |   |   **if** $max(A) > v$ **then**
  |   |   |   |   |   $v \leftarrow max(A)$
  |   |   |   |   **end**
  |   |   |   **end**
  |   |   **end**
  |   **end**
  |   $V[l] \leftarrow v$
**end**

**Algorithm 2** Initialize the neuron parameters. Membrane potential ($U$), last spike time ($S$), dropout mask ($M$). The initialization is performed once for every mini-batch.

---

**Input:** Input($X$), network model($N$)
$b\_size = X.b\_size$
  $h = X.height$
  $w = X.width$
  **for** *l=1* **to** *L* **do**
  |   **if** $isintance(N_l, Conv)$ **then**
  |   |   $U_l = zeros(b\_size, N_l.out, h, w)$
  |   |   $S_l = ones(b\_size, N_l.out, h, w) * (-1000)$
  |   **end**
  |   **else if** $isintance(N_l, Linear)$ **then**
  |   |   $U_l = zeros(b\_size, N_l.out)$
  |   |   $S_l = ones(b\_size, N_l.out) * (-1000)$
  |   **end**
  |   **else if** $isintance(N_l, Dropout)$ **then**
  |   |   // Generate the dropout map that will be fixed for all time steps
  |   |   $M_l = N_l(ones(U_{l-1}.shape))$
  |   **end**
  |   **else if** $isintance(N_l, AvgPool)$ **then**
  |   |   // Reduce the width and height after average pooling layer
  |   |   $h = h // kernel\_size$
  |   |   $w = w // kernel\_size$
  |   **end**
**end**

## 2.2 Spike Timing Dependent Backpropagation (STDB) Learning Rule

The neuron dynamics (Equation 2) show that the neuron's state at a particular time step recurrently depends on its state in previous time steps. This introduces implicit recurrent connections in the network (Neftci et al., 2019). Therefore, the learning rule has to perform the temporal credit assignment along with the spatial credit assignment. Credit assignment refers to the process of assigning credit or blame to the network parameters according to their contribution to the loss function. Spatial credit assignment identifies structural network parameters (like weights), whereas temporal credit assignment determines which past network activities contributed to the loss function. Gradient-descent learning solves both credit assignment problem: spatial credit assignment is performed by distributing error spatially across all layers using the chain rule of derivatives, and temporal credit assignment is done by unrolling the network in time and performing backpropagation through time (BPTT) using the same chain rule of derivatives (Werbos et al., 1990). In BPTT, the network is unrolled for all time steps and the final output is computed as the sum of outputs from each time step. The loss function is defined on the summed output.

The dynamics of the neuron in the output layer is described by Equation (4), where the leak part is removed ($\lambda = 1$) and the neuron only integrates the input without firing. This eliminates the difficulty of defining the loss function on spike count (Lee et al., 2019).

$$u_i^t = u_i^{t-1} + \sum_j w_{ij} o_j \tag{4}$$

The number of neurons in the output layer is the same as the number of categories in the classification task. The output of the network is passed through a softmax layer that outputs a probability distribution. The loss function is defined as the cross-entropy between the true output and the

network's predicted distribution.

$$L = -\sum_i y_i log(p_i) \tag{5}$$

$$p_i = \frac{e^{u_i^T}}{\sum_{k=1}^N e^{u_k^T}} \tag{6}$$

$L$ is the loss function, $y$ the true output, $p$ the prediction, $T$ the total number of time steps, $u^T$ the accumulated membrane potential of the neuron in the output layer from all time steps, and $N$ the number of categories in the task. For deeper networks and large number of time steps the truncated version of the BPTT algorithm is used to avoid memory issues. In the truncated version the loss is computed at some time step $t'$ before T based on the potential accumulated till $t'$. The loss is backpropagated to all layers and the loss gradients are computed and stored. At this point, the history of the computational graph is cleaned to save memory. The subsequent computation of loss gradients at later time steps $(2t', 3t', ...T)$ are summed together with the gradient at $t'$ to get the final gradient. The optimizer updates the parameters at $T$ based on the sum of the gradients. Gradient descent learning has the objective of minimizing the loss function. This is achieved by backpropagating the error and updating the parameters opposite to the direction of the derivative. The derivative of the loss function w.r.t. to the membrane potential of the neuron in the final layer is described by,

$$\frac{\partial L}{\partial u_i^T} = p_i - y_i \tag{7}$$

To compute the gradient at current time step, the membrane potential at last time step ($u_i^{t-1}$ in Equation 4) is considered as an input quantity. Therefore, gradient descent updates the network parameters $W_{ij}$ of the output layer as,

$$W_{ij} = W_{ij} - \eta \Delta W_{ij} \tag{8}$$

$$\Delta W_{ij} = \sum_t \frac{\partial L}{\partial W_{ij}^t} = \sum_t \frac{\partial L}{\partial u_i^T} \frac{\partial u_i^T}{\partial W_{ij}^t} = \frac{\partial L}{\partial u_i^T} \sum_t \frac{\partial u_i^T}{\partial W_{ij}^t} \tag{9}$$

where $\eta$ is the learning rate, and $W_{ij}^t$ represents the copy of the weight used for computation at time step $t$. In the output layer the neurons do not generate a spike, and hence, the issue of non-differentiability is not encountered. The update of the hidden layer parameters is described by,

$$\Delta W_{ij} = \sum_t \frac{\partial L}{\partial W_{ij}^t} = \sum_t \frac{\partial L}{\partial o_i^t} \frac{\partial o_i^t}{\partial u_i^t} \frac{\partial u_i^t}{\partial W_{ij}^t} \tag{10}$$

where $o_i^t$ is the thresholding function (Equation 3) whose derivative w.r.t to $u_i^t$ is zero everywhere and not defined at the time of spike. The challenge of discontinuous spiking nonlinearity is resolved by introducing a surrogate gradient which is the continuous approximation of the real gradient.

$$\frac{\partial o_i^t}{\partial u_i^t} = \alpha e^{-\beta \Delta t} \tag{11}$$

where $\alpha$ and $\beta$ are constants, $\Delta t$ is the time difference between the current time step ($t$) and the last time step the post-neuron generated a spike ($t_s$). It is an integer value whose range is from zero to the total number of time steps ($T$).

$$\Delta t = (t - t_s), 0 < \Delta t < T , \Delta t \, \epsilon \, \mathbb{Z} \tag{12}$$

The values of $\alpha$ and $\beta$ are selected depending on the value of $T$. If $T$ is large $\beta$ is lowered to reduce the exponential decay so a spike can contribute towards gradients for later time steps. The value of $\alpha$ is also reduced for large $T$ because the gradient can propagate through many time steps. The gradient is summed at each time step and thus a large $\alpha$ may lead to exploding gradient. The surrogate gradient can be pre-computed for all values of $\Delta t$ and stored in a look-up table for faster computation. The parameter updates are triggered by the spiking activity but the error gradients are still non-zero for time steps following the spike time. This enables the algorithm to avoid the 'dead

---

**Algorithm 3** Training an SNN with surrogate gradient computed with spike timing. The network is composed of $L$ layers. The training proceeds with mini-batch size ($batch\_size$)

---

**Input:** Mini-batch of input ($X$) - target ($Y$) pairs, network model ($N$), initial weights ($W$), threshold voltage ($V$)

$U, S, M = InitializeNeuronParameters(X)$ [Algorithm 2]
  // Forward propagation
  **for** $t$=1 **to** $T$ **do**
  $\quad O_0^t = PoissonGenerator(X)$
  $\quad$ **for** $l$=1 **to** $L$-1 **do**
  $\quad\quad$ **if** $isintance(N_l, [Conv, Linear])$ **then**
  $\quad\quad\quad$ // accumulate the output of previous layer in $U$, soft reset when spike occurs
  $\quad\quad\quad U_l^t = \lambda U_l^{t-1} + W_l O_{l-1}^t - V_l * O_l^{t-1}$
  $\quad\quad\quad$ // generate the output (+1) if $U$ exceeds $V$
  $\quad\quad\quad O_l^t = STDB(U_l^t, V_l, t)$
  $\quad\quad\quad$ // store the latest spike times for each neuron
  $\quad\quad\quad S_l^t[O_l^t == 1] = t$
  $\quad\quad$ **end**
  $\quad\quad$ **else if** $isintance(N_l, AvgPool)$ **then**
  $\quad\quad\quad O_l^t = N_l(O_{l-1}^t)$
  $\quad\quad$ **end**
  $\quad\quad$ **else if** $isintance(N_l, Dropout)$ **then**
  $\quad\quad\quad O_l^t = O_{l-1}^t * M_l$
  $\quad\quad$ **end**
  $\quad$ **end**
  $\quad U_L^t = \lambda U_L^{t-1} + W_L O_{L-1}^t$
  **end**
// Backward Propagation
  Compute $\frac{\partial L}{\partial U_L}$ from the cross-entropy loss function using BPTT
  **for** $t$=T **to** 1 **do**
  $\quad$ **for** $l$=L-1 **to** 1 **do**
  $\quad\quad$ Compute $\frac{\partial L}{\partial O_l^t}$ based on if $N_l$ is linear, conv, pooling, etc.
  $\quad\quad \frac{\partial L}{\partial U_l^t} = \frac{\partial L}{\partial O_l^t}\frac{\partial O_l^t}{\partial U_l^t} = \frac{\partial L}{\partial O_l^t} * \alpha e^{-\beta S_l^t}$
  $\quad$ **end**
  **end**

---

neuron' problem, where no learning happens when there is no spike. Fig. 1 shows the activation gradient for different values of $\Delta t$, the gradient decreases exponentially for neurons that have not been active for a long time. In Hebbian models of biological learning, the parameter update is activity dependent. This is experimentally observed in spike-timing-dependent plasticity (STDP) learning rule which modulates the weights for pair of neurons that spike within a time window (Song et al., 2000).

## 3 SNN WEIGHT INITIALIZATION

A prevalent method of constructing SNNs for inference is ANN-SNN conversion (Diehl et al., 2015; Sengupta et al., 2019). Since the network is trained with analog activations it does not suffer from the non-differentiablity issue and can leverage the training techniques of ANNs. The conversion process has a major drawback: it suffers from long inference latency ($\sim$2500 time steps) as mentioned in Section 1. As there is no provision to optimize the parameters after conversion based on spiking activity, the network can not leverage the temporal information of the spikes. In this work, we propose to use the conversion process as an initialization technique for STDB. The converted weights and thresholds serve as a good initialization for the optimizer and the STDB learning rule is applied for temporal and spatial credit assignment.

Algorithm 1 explains the ANN-SNN conversion process. The threshold voltages in SNN needs to be adjusted based on the ANN weights. Sengupta et al. (2019) showed two ways to achieve this: weight-normalization and threshold-balancing. In weight-normalization the weights are scaled by a normalization factor and threshold is set to 1, whereas in threshold-balancing the weights are unchanged and the threshold is set to the normalization factor. Both have a similar effect and either can be used to set the threshold. We employ the threshold-balancing method and the normalization factor is calculated as the maximum output of the corresponding convolution/linear layer in SNN. The maximum is calculated over a mini-batch of input for all time steps.

There are several constraints imposed on training the ANN for the conversion process (Sengupta et al., 2019; Diehl et al., 2015). The neurons are trained without the bias term because the bias term in SNN has an indirect effect on the threshold voltage which increases the difficulty of threshold balancing and the process becomes more prone to conversion loss. The absence of bias term eliminates the use of Batch Normalization (Ioffe & Szegedy, 2015) as a regularizer in ANN since it biases the input of each layer to have zero mean. As an alternative, Dropout (Srivastava et al., 2014) is used as a regularizer for both ANN and SNN training. The implementation of Dropout in SNN is further discussed in Section 5. The pooling operation is widely used in ANN to reduce the convolution map size. There are two popular variants: max pooling and average pooling (Boureau et al., 2010). Max (Average) pooling outputs the maximum (average) value in the kernel space of the neuron's activations. In SNN, the activations are binary and performing max pooling will result in significant information loss for the next layer, so we adopt the average pooling for both ANN and SNN (Diehl et al., 2015).

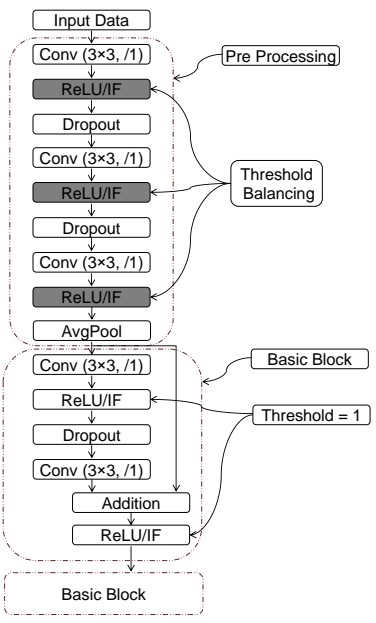

## 4 NETWORK ARCHITECTURES

In this section, we describe the changes made to the VGG (Simonyan & Zisserman, 2014) and residual architecture (He et al., 2016) for hybrid learning and discuss the process of threshold computation for both the architectures.

Figure 2: Residual architecture for SNN

### 4.1 VGG ARCHITECTURE

The threshold balancing is performed for all layers except the input and output layer in a VGG architecture. For every hidden convolution/linear layer the maximum input[3] to the neuron is computed over all time steps and set as threshold for that layer. The threshold assignment is done sequentially as described in Algorithm 1. The threshold computation for all layers can not be performed in parallel (in one forward pass) because in the forward method (Algorithm 3) we need the threshold at each time step to decide if the neuron should spike or not.

### 4.2 RESIDUAL ARCHITECTURE

Residual architectures introduce shortcut connections between layers that are not next to each other. In order to minimize the ANN-SNN conversion loss various considerations were made by Sengupta et al. (2019). The original residual architecture proposed by He et al. (2016) uses an initial convolution layer with wide kernel (7×7, stride 2). For conversion, this is replaced by a pre-processing block consisting of a series of three convolution layer (3×3, stride 1) with dropout layer in between (Fig. 2). The threshold balancing mechanism is applied to only these three layers and the layers in the basic block have unity threshold.

---

[3]input to a neuron is the weighted sum of spkies from pre-neurons $\sum_j w_{ij} o_j$

Table 1: Classification results (Top-1) for CIFAR10, CIFAR100 and ImageNet data sets. Column-1 shows the network architecture. Column-2 shows the ANN accuracy when trained under the constraints as described in Section 3. Column-3 shows the SNN accuracy for $T = 2500$ when converted from a ANN with threshold balancing. Column-4 shows the performance of the same converted SNN with lower time steps and adjusted thresholds. Column-5 shows the performance after training the Column-4 network with STDB for less than 20 epochs.

| Architecture | ANN | ANN-SNN Conversion ($T = 2500$) | ANN-SNN Conversion (reduced time steps) | Hybrid Training (ANN-SNN Conversion + STDB) |
|---|---|---|---|---|
| CIFAR10 | | | | |
| VGG5 | 87.88% | 87.64% | 84.56% ($T = 75$) | 86.91% ($T = 75$) |
| VGG9 | 91.45% | 90.98% | 87.31% ($T = 100$) | 90.54% ($T = 100$) |
| VGG16 | 92.81% | 92.48% | 90.2% ($T = 100$) | 91.13% ($T = 100$) |
| ResNet8 | 91.35% | 91.12% | 89.5% ($T = 200$) | 91.35% ($T = 200$) |
| ResNet20 | 93.15% | 92.94% | 91.12% ($T = 250$) | 92.22% ($T = 250$) |
| CIFAR100 | | | | |
| VGG11 | 71.21% | 70.94% | 65.52% ($T = 125$) | 67.87% ($T = 125$) |
| ImageNet | | | | |
| ResNet34 | 70.2% | 65.1% | 56.87% ($T = 250$) | 61.48% ($T = 250$) |
| VGG16 | 69.35% | 68.12% | 62.73% ($T = 250$) | 65.19% ($T = 250$) |

## 5 OVERALL TRAINING ALGORITHM

Algorithm 1 defines the process to initialize the parameters (weights, thresholds) of SNN based on ANN-SNN conversion. Algorithm 2 and 3 show the mechanism of training the SNN with STDB. Algorithm 2 initializes the neuron parameters for every mini-batch, whereas Algorithm 3 performs the forward and backward propagation and computes the credit assignment. The threshold voltage for all neurons in a layer is same and is not altered in the training process. For each dropout layer we initialize a mask ($M$) for every mini-batch of inputs. The function of dropout is to randomly drop a certain number of inputs in order to avoid overfitting. In case of SNN, inputs are represented as a spike train and we want to keep the dropout units same for the entire duration of the input. Thus, a random mask ($M$) is initialized (Algorithm 2) for every mini-batch and the input is element-wise multiplied with the mask to generate the output of the dropout layer (Lee et al., 2019). The Poisson generator function outputs a Poisson spike train with rate proportional to the pixel value in the input. A random number is generated at every time step for each pixel in the input image. The random number is compared with the normalized pixel value and if the random number is less than the pixel value an output spike is generated. This results in a Poisson spike train with rate equivalent to the pixel value if averaged over a long time. The weighted sum of the input is accumulated in the membrane potential of the first convolution layer. The STDB function compares the membrane potential and the threshold of that layer to generate an output spike. The neurons that output a spike their corresponding entry in $S$ is updated with current time step ($t$). The last spike time is initialized with a large negative number (Algorithm 2) to denote that at the beginning the last spike happened at negative infinity time. This is repeated for all layers until the last layer. For last layer the inputs are accumulated over all time steps and passed through a softmax layer to compute the multi-class probability. The cross-entropy loss function is defined on the output of the softmax and the weights are updated by performing the temporal and spatial credit assignment according to the STDB rule.

## 6 EXPERIMENTS

We tested the proposed training mechanism on image classification tasks from CIFAR (Krizhevsky et al., 2009) and ImageNet (Deng et al., 2009) datasets. The results are summarized in Table 1.

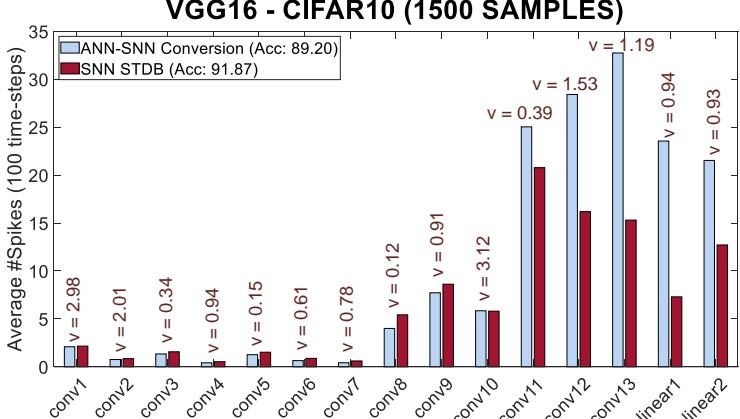

Figure 3: Average number of spikes for each layer in a VGG16 architecture for purely converted SNN and SNN trained with hybrid technique. The converted SNN and SNN trained with hybrid technique achieve an accuracy of 89.20% and 91.87%, respectively, for the randomly selected 1500 samples from the test set. Both the networks were inferred for 100 time steps and 'v' represents the threshold voltage for each layer obtained during the conversion process (Algorithm 1).

CIFAR10: The dataset consists of labeled $60,000$ images of 10 categories divided into training ($50,000$) and testing ($10,000$) set. The images are of size $32 \times 32$ with RGB channels.

CIFAR100: The dataset is similar to CIFAR10 except that it has 100 categories.

ImageNet: The dataset comprises of labeled high-resolution 1.2 million training images and $50,000$ validation images with 1000 categories.

## 7  ENERGY-DELAY PRODUCT ANALYSIS OF SNNS

A single spike in an SNN consumes a constant amount of energy (Cao et al., 2015). The first order analysis of energy-delay product of an SNN is dependent on the number of spikes and the total number of time steps. Fig. 3 shows the average number of spikes in each layer when evaluated for 1500 samples from CIFAR10 testset for VGG16 architecture. The average is computed by summing all the spikes in a layer over 100 time steps and dividing by the number of neurons in that layer. For example, the average number of spikes in the $10^{th}$ layer is 5.8 for both the networks, which implies that over a 100 time step period each neuron in that layer spikes 5.8 times on average over all input samples. Higher spiking activity corresponds to lower energy-efficiency. The average number of spikes is compared for a converted SNN and SNN trained with conversion-and-STDB. The SNN trained with conversion-and-STDB has $1.5\times$ less number of average spikes over all layers under iso conditions (time steps, threshold voltages, inputs, etc.) and achieves higher accuracy compared to the converted SNN. The converted SNNs when simulated for larger time steps further degrade the energy-delay product with minimal increase in accuracy (Sengupta et al., 2019).

## 8  RELATED WORK

Bohte et al. (2000) proposed a method to directly train on SNN by keeping track of the membrane potential of spiking neurons only at spike times and backpropagating the error at spike times based on only the membrane potential. This method is not suitable for networks with sparse activity due to the 'dead neuron' problem: no learning happens when the neurons do not spike. In our work, we need one spike for the learning to start but gradient contribution continues in later time steps as shown in Fig. 1. Zenke & Ganguli (2018) derived a surrogate gradient based method on the membrane potential of a spiking neuron at a single time step only. The error was backpropagated at only one time step and only the input at that time step contributed to the gradient. This method neglects the effect of earlier spike inputs. In our approach, the error is backpropagated for every time step and the weight update is performed on the gradients summed over all time steps. Shrestha & Orchard (2018) proposed a gradient function similar to the one proposed in this work. They

Table 2: Comparion of our work with other SNN models on CIFAR10 and ImageNet datasets

| Model | Dataset | Training Method | Architecture | Accuracy | Time-steps |
|---|---|---|---|---|---|
| Hunsberger & Eliasmith (2015) | CIFAR10 | ANN-SNN Conversion | 2Conv, 2Linear | 82.95% | 6000 |
| Cao et al. (2015) | CIFAR10 | ANN-SNN Conversion | 3Conv, 2Linear | 77.43% | 400 |
| Sengupta et al. (2019) | CIFAR10 | ANN-SNN Conversion | VGG16 | 91.55% | 2500 |
| Lee et al. (2019) | CIFAR10 | Spiking BP | VGG9 | 90.45% | 100 |
| Wu et al. (2019) | CIFAR10 | Surrogate Gradient | 5Conv, 2Linear | 90.53% | 12 |
| **This work** | **CIFAR10** | **Hybrid Training** | **VGG16** | **91.13% 92.02%** | **100 200** |
| Sengupta et al. (2019) | ImageNet | ANN-SNN Conversion | VGG16 | 69.96% | 2500 |
| **This work** | **ImageNet** | **Hybrid Training** | **VGG16** | **65.19%** | **250** |

used the difference between the membrane potential and the threshold to compute the gradient compared to the difference in spike timing used in this work. The membrane potential is a continuous value whereas the spike time is an integer value bounded by the number of time steps. Therefore, gradients that depend on spike time can be pre-computed and stored in a look-up table for faster computation. They evaluated their approach on shallow architectures with two convolution layer for MNIST dataset. In this work, we trained deep SNNs with multiple stacked layers for complex calssification tasks. Wu et al. (2018) performed backpropagation through time on SNN with a surrogate gradient defined on the membrane potential. The surrogate gradient was defined as piece-wise linear or exponential function of the membrane potential. The other surrogate gradients proposed in the literature are all computed on the membrane potential (Neftci et al., 2019). Lee et al. (2019) approximated the neuron output as continuous low-pass filtered spike train. They used this approximated continuous value to perform backpropagation. Most of the works in the literature on direct training of SNN or conversion based methods have been evaluated on shallow architectures for simple classification problems. In Table 2 we compare our model with the models that reported accuracy on CIFAR10 and ImageNet dataset. Wu et al. (2019) achieved convergence in 12 time steps by using a dedicated encoding layer to capture the input precision. It is beyond the scope of this work to compute the hardware and energy implications of such encoding layer. Our model performs better than all other models at far fewer number of time steps.

## 9 CONCLUSIONS

The direct training of SNN with backpropagation is computationally expensive and slow, whereas ANN-SNN conversion suffers from high latency. To address this issue we proposed a hybrid training technique for deep SNNs. We took an SNN converted from ANN and used its weights and thresholds as initialization for spike-based backpropagation of SNN. We then performed spike-based backpropagation on this initialized network to obtain an SNN that can perform with fewer number of time steps. The number of epochs required to train SNN was also reduced by having a good initial starting point. The resultant trained SNN had higher accuracy and lower number of spikes/inference compared to purely converted SNNs at reduced number of time steps. The backpropagation through time was performed with surrogate gradient defined using neuron's spike time that captured the temporal information and helped in reducing the number of time steps. We tested our algorithm on CIFAR and ImageNet datasets and achieved state-of-the-art performance with fewer number of time steps.

ACKNOWLEDGMENTS

This work was supported in part by the National Science Foundation, in part by Vannevar Bush Faculty Fellowship, and in part by C-BRIC, one of six centers in JUMP, a Semiconductor Research Corporation (SRC) program sponsored by DARPA.

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

## A    COMPARISONS WITH OTHER SURROGATE GRADIENTS

The transfer function of the spiking neuron is a step function and its derivative is zero everywhere except at the time of spike where it is not defined. In order to perform backpropagation with spiking neuron several approximations are proposed for the gradient function (Bellec et al., 2018; Zenke & Ganguli, 2018; Shrestha & Orchard, 2018; Wu et al., 2018). These approximations are either a linear or exponential function of $(u - V_t)$, where $u$ is the membrane potential and $V_t$ the threshold voltage (Fig. 4). These approximations are referred as surrogate gradient or pseudo-derivative. In this work, we proposed an approximation that is computed using the spike timing of the neuron (Equation 11). We compare our proposed approximation with the following surrogate gradients:

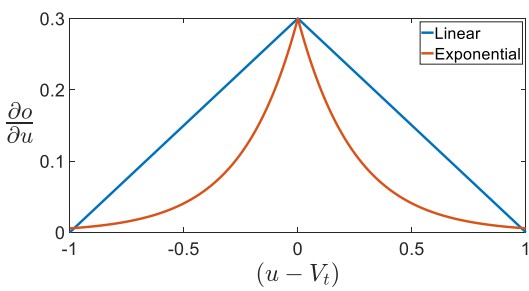

Figure 4: Linear and Exponential approximation of the gradient of the spiking neuron (step function).

$$\frac{\partial o}{\partial u} = \alpha \, max\{0, 1 - |u - V_t|\} \tag{13}$$

$$\frac{\partial o}{\partial u} = \alpha e^{-\beta \, |u - V_t|} \tag{14}$$

where $o$ is the binary output of the neuron, $u$ is the membrane potential, $V_t$ is the threshold potential, $\alpha$ and $\beta$ are constants. Equation 13 and Equation 14 represent the linear and exponential approximation of the gradient, respectively. We employed these approximations in the hybrid training for a VGG9 network for CIFAR10 dataset. All the approximations (Equation 11, 13, and 14) produced similar results in terms of accuracy and number of epochs for convergence. This shows that the term $\Delta t$ (Equation 11) is a good replacement for $|u - V_t|$ (Equation 14). The behaviour of $\Delta t$ and $|u - V_t|$ is similar, i.e., it is small closer to the time of spike and increases as we move away from the spiking event. The advantage of using $\Delta t$ is that its domain is bounded by the total number of time steps (Equation 12). Hence, all possible values of gradients can be pre-computed and stored in a table for faster access during training. This is not possible for membrane potential because it is a real value computed based on the stochastic inputs and previous state of the neuron which is not known before hand. The exact benefit in energy from the pre-computation is dependent on the overall system architecture and evaluating it is beyond the scope of this paper.

## B  COMPARISONS OF SIMULATION TIME AND MEMORY REQUIREMENTS

The simulation time and memory requirements for ANN and SNN are very different. SNN requires much more resources to iterate over multiple time steps and store the membrane potential for each neuron. Fig. 5 shows the training and inference time and memory requirements for ANN, SNN trained with backpropagation from scratch, and SNN trained with the proposed hybrid technique. The performance was evaluated for VGG16 architecture trained for CIFAR10 dataset. SNN trained from scratch and SNN trained with hybrid conversion-and-STDB are evaluated for 100 time steps. One epoch of ANN training (inference) takes 0.57 (0.05) minutes and 1.47 (1.15) GB of GPU memory. On the other hand, one epoch of SNN training (inference) takes 78 (11.39) minutes and 9.36 (1.37) GB of GPU memory for same hardware and mini-batch size. ANN and SNN trained from scratch reached convergence after 250 epochs. The hybrid technique requires 250 epochs of ANN training and 20 epochs of spike-based backpropagation. The hybrid training technique is one order of magnitude faster than training SNN from scratch. The memory requirement for hybrid technique is same as SNN as we need to perform fine-tuning with spike-based backpropagation.

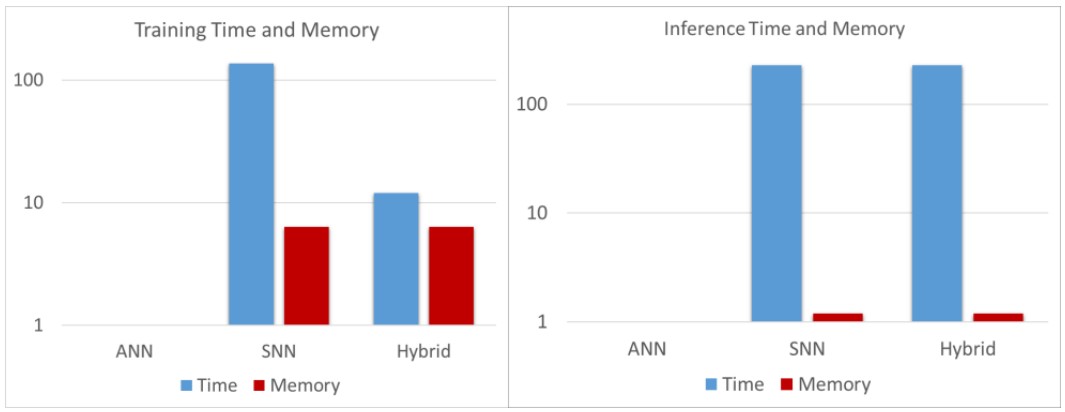

Figure 5: Training and Inference time and memory for ANN, SNN trained with backpropagation from scratch, and SNN trained with hybrid technique. All values are normalized based on ANN values. The y-axis is in log scale. The performance was evaluated on one Nvidia GeForce RTX 2080 Ti TU102 GPU with 11 GB of memory. All the networks were trained for VGG16 architecture, CIFAR10 dataset, 100 time steps, and mini-batch size of 32. ANN and SNN require 250 epochs of training from scratch, hybrid conversion-and-STDB based training requires 250 epochs of ANN training followed by 20 epochs of spike-based backpropagation.

