# OpenReview forum: "Enabling Deep Spiking Neural Networks with Hybrid Conversion and Spike Timing Dependent Backpropagation"
_ICLR.cc/2020/Conference — Accept (Poster)_

### Official Review · AnonReviewer2 · 2019-10-23
**Official Blind Review #2**

**Rating:** 6

**Review:**

This paper presents a fine-tuning method of models converted from standard encoding and SGD training to Spike/NN's.
The key point of the paper is that directly training S/NN's with spike-back-prop is slow and inefficient, while directly inferencing with converted models is also inefficient due to the large integration window required to get a good estimate of the spiking neuron potential. The authors claim, and to a good extent show that, their proposed method is best of both worlds: train the models efficiently with standard encoding / SGD, this is something we know works and scale well, then convert and fine-tune with spike-backprop to get models that perform well under a shorter integration window, and thus are more efficient at inference time. The intuition is that models can achieve shorter integration windows while keeping good results because, under the assumptions made by the proposed algorithm, the fine-tuning is effectively unrolling neuron dynamics that can be trained with BPPT, in a way similar to LSTM/Recurrent models. In that case, since model dynamics are taken into account during fine-tuning, it results in better performance even under shorter time-windows. This is an interesting concept, since the training doesn't only consider a mean-field estimate of the spike-activation, but it looks at  spiking neuron dynamics with an higher granularity. The paper is well written, clear and easy to understand. Results are comparatively competitive and code is made available.

**Experience Assessment:**

I have read many papers in this area.

**Review Assessment: Checking Correctness Of Derivations And Theory:**

I assessed the sensibility of the derivations and theory.

**Review Assessment: Checking Correctness Of Experiments:**

I carefully checked the experiments.

**Review Assessment: Thoroughness In Paper Reading:**

I read the paper at least twice and used my best judgement in assessing the paper.

---

> ### Author Response · Authors · 2019-11-14
> **Response to Review #2**
>
> Thank you for your remarks. You have summarized our paper very nicely and we appreciate your time and effort for reviewing our work. Please let us know if you have any suggestions that can help to increase the score of the paper. We have added Appendix A and B to compare different pseudo-derivatives and the training effort for training an SNN from scratch compared to hybrid conversion-and-STDB training.

---

### Official Review · AnonReviewer3 · 2019-10-23
**Official Blind Review #3**

**Rating:** 6

**Review:**

This paper proposes methods to initialize and train spiking NNs (SNNs) as an alternative to ANNs, not driven primarily by improved loss or generalization, but by energy efficiency improvements derived from timing-event based sparse operation rather than asynchronous sweeps. The backpropagation method introduced is sensible, as are the experiments on known datasets to show its effectiveness. The paper is well written (apart from the miniscule Figure 3 containing the main result).
I recommend acceptance, with caveats: the energy performance is actually not directly calculated, but speculatively estimated, it depends on the computational architecture chosen to implement the respective networks. I point out that ANNs need to be trained first to properly initialize an SNN, so the relative training effort claimed is less impressive, but energy performance does count in actual operational practice - training is (or should) be a small fraction of that.

**Experience Assessment:**

I have read many papers in this area.

**Review Assessment: Checking Correctness Of Derivations And Theory:**

I assessed the sensibility of the derivations and theory.

**Review Assessment: Checking Correctness Of Experiments:**

I assessed the sensibility of the experiments.

**Review Assessment: Thoroughness In Paper Reading:**

I read the paper at least twice and used my best judgement in assessing the paper.

---

> ### Author Response · Authors · 2019-11-14
> **Response to Review #3**
>
> We thank the reviewer for their time and effort to review our work. Please find below the explanation for individual comments.
>
> 1. Reviewer's comment: This paper proposes methods to initialize and train spiking NNs (SNNs) as an alternative to ANNs, not driven primarily by improved loss or generalization, but by energy efficiency improvements derived from timing-event based sparse operation rather than asynchronous sweeps. The backpropagation method introduced is sensible, as are the experiments on known datasets to show its effectiveness.
> The paper is well written (apart from the miniscule Figure 3 containing the main result).
>
> Author's response:  Thank you for pointing this out. We have edited Figure 3.
>
> 2. Reviewer's comment: I recommend acceptance, with caveats: the energy performance is actually not directly calculated, but speculatively estimated, it depends on the computational architecture chosen to implement the respective networks. I point out that ANNs need to be trained first to properly initialize an SNN, so the relative training effort claimed is less impressive, but energy performance does count in actual operational practice - training is (or should) be a small fraction of that.
>
> Author's response: We absolutely agree with the reviewer that the energy estimate is dependent on the network architecture as the number of spikes will vary with architecture. We compute the efficiency of the network in terms of latency (number of time-steps) and the number of spikes per image during inference. We have added Appendix B to compare the training and testing effort for both ANN and SNN.

---

### Official Review · AnonReviewer4 · 2019-11-07
**Official Blind Review #4**

**Rating:** 6

**Review:**

This paper examines combining two approaches of obtaining a trained spikingneural network (SNN). The first approach of previous work is converting the weights of a trained artificial neural network (ANN) with a given architecture, to the weights and thresholds of a SNN, and the second approach uses a surrogate gradient to train an SNN with backpropagation. The true novelty of the paper seems to be in showing that combining the two approaches sequentially, trains a SNN that requiresfewer timesteps to determine an output which achieves state of the art performance. This is summarized by Table 1. However, it does not mention how many epochs it takes to train an SNN from scratch, nor compare this to the total training time (ANN training + SNN fine-tuning) of their approach. They also claim a novel spike-time based surrogate gradient function (eq. 11), but it is very practicallysimilar to the ones explored in the referenced Wu. et al 2018 (eq. 27 for instance), and these should be properly contrasted showing that this novel surrogate function is actually helpful (the performance/energy efficiency might only come from the hybrid approach). The authors argue for SOTA performance in Table 2, but the comparison to other work doesn’t clearly separate their performance from the otherlisted works; For example the accuracy gain against Lee et al.,2019 only comes from the architecture being VGG16 as opposed to VGG9, as can be seen from comparing with the VGG9 architecture from Table 1, furthermore they take the sameamount of timesteps, which is supposed to be the principle gain of this work.

Some small suggestions that are independent from the above:

1.The most similar or relevant version of equation (2) in previous work could be referenced nearby for context.

2.The last sentence of the first paragraph on p.4 “the outputs from each copy...” is confusing. Are you just meaning to describe BPTT?

3.Typos: sec7 4th line “neruons”, sec 2.2 “both the credit” (remove “the”)

---------------
Following the author response I have upgraded my rating.

**Experience Assessment:**

I have read many papers in this area.

**Review Assessment: Checking Correctness Of Derivations And Theory:**

N/A

**Review Assessment: Checking Correctness Of Experiments:**

I assessed the sensibility of the experiments.

**Review Assessment: Thoroughness In Paper Reading:**

I read the paper at least twice and used my best judgement in assessing the paper.

---

> ### Author Response · Authors · 2019-11-14
> **Response to Review #4 Part 1/2**
>
> We thank the reviewer for the detailed comments. Please find below the explanation for individual comments.
>
> 1. Reviewer's comment: "This paper examines combining two approaches of obtaining a trained spiking neural network (SNN). The first approach of previous work is converting the weights of a trained artificial neural network (ANN) with a given architecture, to the weights and thresholds of a SNN, and the second approach uses a surrogate gradient to train an SNN with backpropagation. The true novelty of the paper seems to be in showing that combining the two approaches sequentially, trains a SNN that requires fewer timesteps to determine an output which achieves state of the art performance. This is summarized by Table 1."
>
> Author's response: The proposed hybrid technique trains an SNN that can process the input information in fewer time-steps compared to purely ANN-SNN conversion methods as well as with this technique the training effort in the spiking domain is reduced. Backpropagation training with spikes is expensive because multiple iterations are performed in the forward pass and it requires larger memory to store all the activations for backpropagation. The training effort for ANN and SNN is compared in Appendix B of the revised manuscript. The high training effort for SNN hindered its application for large and complex datasets like ImageNet. The hybrid approach makes it possible to train SNNs for large datasets with a reasonable amount of time and memory by having a good initialization and only fine-tuning in spiking domain for few epochs to reduce the number of time-steps.
>
> 2. Reviewer's comment: "However, it does not mention how many epochs it takes to train an SNN from scratch, nor compare this to the total training time (ANN training + SNN fine-tuning) of their approach".
>
> Author's response: Thank you for pointing this out. We performed simulations to quantify the effort of training SNN from scratch compared to the hybrid training technique with similar hardware constraints. The results are shown in Appendix B of the revised manuscript.
>
> 3. Reviewer's comment: "They also claim a novel spike-time based surrogate gradient function (eq. 11), but it is very practically similar to the ones explored in the referenced Wu. et al 2018 (eq. 27 for instance), and these should be properly contrasted showing that this novel surrogate function is actually helpful (the performance/energy efficiency might only come from the hybrid approach)".
>
> Author's response: Thank you for pointing this out. Wu et al. (2018) mentioned several approximations for the derivative of the spike function. Similar approximations have been proposed in other works as well (Bellec et al., 2018; Zenke & Ganguli 2018; Shrestha & Orchard, 2018). These approximations are either a linear or exponential function of (u-Vt) where u is the membrane potential and Vt the threshold voltage. In our work, we propose to replace the quantity (u- Vt) with Δt, where Δt is the time from the last spike. Δt and (u- Vt) have similar behavior, i.e., their value is small close to the spike event and the value increases as we move away from the spike event. Therefore, Δt works as a good replacement for (u- Vt) and it may lead to energy/computation benefits. All possible values of Δt is bounded and known once the number of time-steps is fixed (Equation 12), therefore we can pre-compute the gradient values and store it in a look-up table for faster access during backpropagation. On the other hand, (u- Vt) is a dynamically changing real number and the gradient must be computed during the simulation. The gradient is computed at every time-step for BPTT and the look-up table may provide faster access compared to computing the gradient based on (u- Vt). The exact benefit in energy is dependent on the overall system architecture and evaluating it is beyond the scope of this paper. We tested our hybrid technique with other approximations based on (u- Vt) and achieved similar performance. The proposed hybrid technique is a general methodology for training deep SNNs and any existing spike-based backpropagation mechanism can be used in this technique. We have added Appendix A to compare various approximations of the gradient of spike function.

---

> > ### Author Response · Authors · 2019-11-14
> > **Response to Review #4 Part 2/2**
> >
> > 4. Reviewer's comment: "The authors argue for SOTA performance in Table 2, but the comparison to other work doesn’t clearly separate their performance from the other listed works; For example the accuracy gain against Lee et al.,2019 only comes from the architecture being VGG16 as opposed to VGG9, as can be seen from comparing with the VGG9 architecture from Table 1, furthermore they take the same amount of timesteps, which is supposed to be the principle gain of this work".
> >
> > Author's response: Thank you for mentioning this. In Table 2, we compare the best results (over all architectures) reported in various works and compare it with our best results for same datasets. As correctly pointed out by the reviewer, the hybrid approach performs at par (for similar architecture) with the results reported by Lee et al. (2019). Lee et al. (2019) employed a backpropagation mechanism to train the SNN from scratch. As we show in Appendix B, the effort for training SNN from scratch is 10X more compared to hybrid training. Therefore, for most practical purposes it is not possible to train deeper SNNs from scratch. In our approach, we start with a network that performs well with large number of time-steps and train the network with spike-based BPTT to reduce the number of inference time-steps. The entire goal of training in spiking regime is to reduce the number of time-steps and therefore it is very natural to achieve similar performance if the same network is trained from scratch with same number of time-steps. The benefit we achieve with hybrid training is convergence within few epochs because we start with a good initialization and therefore this technique can scale to deeper networks and larger datasets.
> >
> > 5. Reviewer's comment: "Some small suggestions that are independent from the above: 1. The most similar or relevant version of equation (2) in previous work could be referenced nearby for context".
> >
> > Author's response: We have added a reference for the iterative modeling of the LIF neuron in Section 2.1
> >
> > 6. Reviewer's comment: "The last sentence of the first paragraph on p.4 “the outputs from each copy...” is confusing. Are you just meaning to describe BPTT?"
> >
> > Author's response: We have edited the sentence to be more concise and clearer. Yes, we are referring to the BPTT mechanism.
> >
> > 7. Reviewer's comment: "Typos: sec7 4th line “neruons”, sec 2.2 “both the credit” (remove “the”)"
> >
> > Author's response:  Thank you for pointing these out. We have made the corrections in the paper.

---

### Decision · Program_Chairs · 2019-12-19

**Decision:**

Accept (Poster)

**Comment:**

After the rebuttal, all reviewers rated this paper as a weak accept.
The reviewer leaning towards rejection was satisfied with the author response and ended up raising their rating to a weak accept.  The AC recommends acceptance.